# An alternative form of the super-Gaussian wind turbine wake model

Frédéric Blondel and Marie Cathelain

IFP Énergies nouvelles, 1&4 Avenue du Bois Préau, 92862 Rueil-Malmaison, France

**Correspondence:** Frédéric Blondel (frederic.blondel@ifpen.fr)

**Abstract.** A new analytical wind turbine wake model, based on a super-Gaussian shape function, is presented. The super-Gaussian function evolves from a nearly top-hat shape in the near wake to a Gaussian shape in the far wake, which is consistent with observations and measurements of wind turbine wakes. Using such a shape function allows to recover the mass and momentum conservation that is violated when applying a near-wake regularization function to the expression of the maximum velocity deficit of the Gaussian wake model. After a brief introduction of the theoretical aspects, an easy-to-implement model with a limited number of parameters is derived. The super-Gaussian model predictions are compared to wind tunnel measurements, full-scale measurements and a large-eddy simulation, showing a good agreement and an improvement compared with predictions based on the Gaussian model.

## 1 Introduction

During the design phase of a wind farm, wind turbine positions must be carefully chosen in order to maximize the power output and minimize the wake effects within a given geographical area. Indeed, even offshore, the area can be limited by several constraints, such as sea bed type (presence of sand banks), national borders, fishing areas, etc. Furthermore, in a wind farm, wind turbines may operate in the wake of upwind turbines. Wind turbine wakes are characterized by a reduction of the wind velocity and an increase of the turbulence level. In the near wake, i.e. at a distance below four wind turbine diameters, the decrease in wind velocity is very strong. In the far wake, at a distance greater than four wind turbine diameters, turbulent mixing leads to wake recovery. Thus, short separation distances between wind turbines lead to higher wake losses. In the end, a complex optimization problem, implying a large number of evaluations of the wind farm power, must be solved to maximize the wind energy production (or minimize wake losses) on a given site with given wind characteristics. Despite the availability of high-fidelity methods (Churchfield et al. (2012), Joulin et al. (2019)), wind farm designs are still based on analytical wake models, because they are computationally affordable.

Several analytical models have been derived over the years, from the well-known work of Jensen (1983) and Katic et al. (1987) to the most recent models proposed by Frandsen et al. (2006), or Bastankhah and Porté-Agel (2014). These models are designed to estimate the far-wake characteristics. However, wind turbine separation distances in wind farms can be small, i.e. below four wind turbine diameters ($d_0$). A typical example is the Lillgrund wind farm, with a minimal separation distance of $3.3d_0$. Thus, analytical models should be accurate not only in the far wake, but also in the near wake.

In the work of Frandsen et al. (2006) and Bastankhah and Porté-Agel (2014), two closely related shortcomings have to be alleviated: firstly, due to the choice of a Gaussian shape and from mass and momentum conservation, the maximum velocity deficit decreases with the distance to the rotor. This is not exact in the near wake: the velocity deficit increases, reaches a maximum value, and then decreases, due to the turbulent mixing. Secondly, it has been observed, both numerically and experimentally, that the wake velocity profiles are not purely Gaussian, as supposed in the aforementioned model, but evolve downstream the wind turbine from a top-hat shape to a Gaussian shape in the far wake (see Lissaman (1979), Aubrun et al. (2013), Sørensen et al. (2015), Bartl and Sætran (2017)). This is due to the tip vortices, that emanate from the blade tips and break up while propagating downstream the wind turbine, leading to a mixing of the wake with the atmospheric flow and finally to the Gaussian-shaped velocity profile. Having a correct wake shape is fundamental, since a wrong shape can lead to erroneous power estimation for a rotor operating in full-wake or partial-wake conditions.

Recently, Qian and Ishihara (2018) proposed a modified version of the Bastankhah and Porté-Agel model that improves the velocity deficit prediction in the near wake. In this updated model, a corrective term is added in order to predict realistic near-wake velocities. However, by using such a corrective term, mass and momentum conservation is violated.

Another velocity deficit distribution has been proposed by Keane et al. (2016) and Schreiber et al. (2020): their model is derived by applying conservation of mass and momentum in the context of actuator disk theory, but assumes a distribution of the double-Gaussian type for the velocity deficit in the wake. Indeed, the authors consider that the near wake is better approximated using a double-Gaussian distribution. However, other experiments by Bastankhah and Porté-Agel (2020) show an increase of the velocity deficit at the wake center in the very near wake ($< 2d_0$) even so they were not designed for measuring the very near wake. It has been decided to neglect the misunderstood effect of the nacelle as it is supposed to vanish after $2d_0$ and to propose a generic formulation of the super-Gaussian type. This specific effect could be included as a correction added to the present formulation of the model. Comparisons between the proposed model and the double-Gaussian wake model are not straightforward, since, to our best knowledge, no "generic" formulation (or calibration) of the double-Gaussian wake model has been proposed yet (i.e. no dependence on thrust coefficient or turbulence intensity are considered in the two aforementioned references). The double-Gaussian wake model has therefore been excluded from the present comparisons.

In the present work, it is shown that by using a super-Gaussian shape, the wake velocity profiles are more consistent with observations, the velocity deficit has the expected form, and mass and momentum conservation is preserved. Indeed, the super-Gaussian function tends towards a top-hat shape for high values of the super-Gaussian order $n$ (near-wake conditions), while for $n = 2$, the traditional Gaussian shape is recovered (far-wake conditions). In the near wake, the top-hat shape can be altered by the presence of the hub and tower wakes, or even by the non-uniform distribution of the inductions on the blade (as observed during the Mexico and NewMexico experimental campaigns, see Boorsma et al. (2019)). These effects are neglected in the present work, since they tend to be rapidly dissipated (one to two diameters behind the wind turbine), and the wake rapidly transitions towards a smooth top-hat shape before turbulent mixing takes place and leads to the well-known Gaussian shape in the far wake.

The idea of using a super-Gaussian shape function has already been suggested in Shapiro et al. (2019). In the present work, an alternative formulation is presented. Both mass and momentum are conserved, following the derivation of the Gaussian

model of Bastankhah and Porté-Agel (2014), whereas in the work of Shapiro et al. (2019), only mass conservation is enforced. Furthermore, a new form of the near-wake correction for the velocity deficit proposed by Qian and Ishihara (2018) is presented, and an analytical expression for the evolution of the super-Gaussian order $n$ as a function of the downstream distance is proposed. This expression is obtained by enforcing mass and momentum conservation, using the aforementioned near-wake corrected velocity deficit model and assuming a linear evolution of the wake width with respect to the downstream distance.

Finally, the model is calibrated on a wide range of thrust coefficients and turbulence intensities by using measured velocity profiles behind an actuator disk (Aubrun et al. (2013) and Sumner et al. (2013)) and an onshore wind turbine (Doubrawa et al. (2019)). Results of a Large Eddy Simulation (LES) are also used as a reference for the wind turbine case. Comparison with experimental velocity profiles highlights the improvement brought by the super-Gaussian model over the Gaussian model.

## 2 The super-Gaussian wake model

### 2.1 Model derivation

The derivation of the super-Gaussian wake model closely follows the one proposed by Bastankhah and Porté-Agel (2014). The non-dimensional velocity deficit in the wake is expressed as the product of the maximum velocity deficit $C(\tilde{x})$ and a shape function $f(\tilde{r})$, with $\tilde{x}$, $\tilde{\sigma}$ and $\tilde{r}$ the axial distance from the turbine, the characteristic wake width (which is the standard deviation when $n = 2$) and the radial distance from the wake center, all three normalized by the wind turbine diameter, $d_0$:

$$\frac{U_\infty - U_w}{U_\infty} = C(\tilde{x})f(\tilde{r}) = C(\tilde{x})e^{-\tilde{r}^n/(2\tilde{\sigma}^2)}, \tag{1}$$

with $U_\infty$ the wind velocity at infinity and $U_w$ the velocity in the wake. In the rest of the document, the tilde symbol denotes a normalization by the wind turbine diameter, $d_0$. Furthermore, the dependence on $\tilde{x}$ for $\tilde{\sigma}(\tilde{x})$ and $\tilde{n}(\tilde{x})$ is omitted to simplify the notations: $\tilde{\sigma} = \tilde{\sigma}(\tilde{x})$ and $\tilde{n} = \tilde{n}(\tilde{x})$. The shape function, $f(\tilde{r})$, takes a form similar to a super-Gaussian function, with a squared characteristic wake width $\tilde{\sigma}$. The characteristic wake width is directly linked to the wake width. The super-Gaussian function is a convenient choice for representing wakes, since for high values of the super-Gaussian order $n$, the function is close to a top-hat, as observed in the near wake, while for lower values of $n$, the function smoothly evolves towards the well-known Gaussian shape, as observed in the far wake. For $n = 2$, the super-Gaussian is actually a Gaussian function.

Typical super-Gaussian profiles are shown in Fig. 1. Depending on the characteristic wake width $\tilde{\sigma}$, the wake width at the base can be slightly larger or thinner compared with the Gaussian counterpart ($n = 2$). The highest value of characteristic wake width ($\tilde{\sigma} = 0.6$), for which the wake base is thinner with the super-Gaussian model, is typical of far wake and high turbulence conditions. This case is not likely to occur, since a Gaussian shape is expected in the far wake.

The model is derived by enforcing mass and momentum conservation. Only the main results are given here. Detailed calculations can be found in Appendix A. According to Frandsen et al. (2006), applying mass and momentum conservation leads to the following equation:

$$2\pi\rho \int\limits_0^\infty U_w \left(U_\infty - U_w\right) r \, dr = T, \tag{2}$$

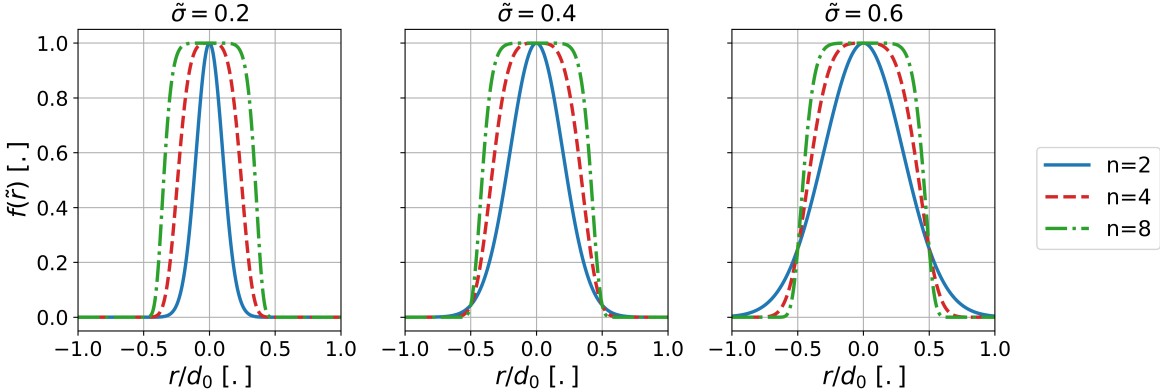

**Figure 1.** Super-Gaussian profiles of orders $n = 2$ to $n = 8$ for three different characteristic wake width values.

where $\rho$ is the air density, and $T$ is the thrust force applied by the wind turbine on the flow. This force is related to the thrust coefficient $C_T$, which is a manufacturer data supposed to be known, and function of the infinite wind velocity. With $A_0$ the rotor swept area, the thrust writes:

$$T = \frac{1}{2}\rho C_T A_0 U_\infty^2.$$
(3)

As shown by Eq. (2) and (3), the thrust coefficient is a non-dimensional variable, that represents the loss of kinetic energy of the flow due to the presence of the wind turbine. It scales the intensity of the velocity deficit in the wake. After inserting Eq. (1) into Eq. (2), the following relation is obtained, with $\Gamma$ the Gamma function:

$$C(\tilde{x})^2 - 2^{2/n}C(\tilde{x}) + n\frac{C_T}{16\Gamma(2/n)\tilde{\sigma}^{4/n}} = 0.$$
(4)

From Eq. (4), it is possible to derive an expression for the maximum velocity deficit:

$$C(\tilde{x}) = 2^{2/n-1} - \sqrt{2^{4/n-2} - \frac{nC_T}{16\Gamma(2/n)\tilde{\sigma}^{4/n}}}.$$
(5)

The original form of $C(\tilde{x})$ proposed by Bastankhah and Porté-Agel is recovered (with $\Gamma(1) = 1$) when setting the super-Gaussian order to $n = 2$:

$$C(\tilde{x}) = 1 - \sqrt{1 - \frac{C_T}{8\tilde{\sigma}^2}}.$$
(6)

## 2.2   Model implementation

### 2.2.1   Root-finding approach

In the Gaussian model, there are only two unknown variables, the normalized characteristic wake width $\tilde{\sigma}$ and the maximum velocity deficit $C(\tilde{x})$. The maximum velocity deficit can be obtained by using a linear evolution of the characteristic wake

width with respect to the distance to the rotor, see Eq. (9). The linear assumption is based on the analysis of experimental and numerical data. In the super-Gaussian formulation, another variable is introduced: the super-Gaussian order $n$. A first idea is to keep the linear assumption for the wake characteristic width, set the super-Gaussian order $n$ to get the desired wake shape, and calculate the maximum velocity deficit. Using such a method, an expression for $n$ needs to be found.

Here, a different approach is used. A linear evolution of the characteristic wake width is considered. Then, the maximum velocity deficit is calculated using the model of Bastankhah and Porté-Agel (2014), augmented with a near-wake correction similar to the one introduced by Qian and Ishihara (2018). As already mentioned, by using a Gaussian wake model, the introduction of this near-wake correction violates the mass and momentum conservation. Once the super-Gaussian shape function is introduced, mass and momentum conservation can be preserved by choosing $n$ accordingly (i.e. using Eq. (4)). Finally, the velocity in the wake can be computed using Eq. (1).

Explicit forms of the near-wake correction, characteristic wake width and maximum velocity deficit need to be defined. The near-wake correction, denoted $\kappa(\tilde{x})$, takes the following form:

$$\kappa(\tilde{x}) = c_{NW} \left(1 + \tilde{x}\right)^{p_{NW}},$$ (7)

with $c_{NW}$ and $p_{NW}$ two parameters of the correction. Introducing Eq. (7) in the expression of the velocity deficit proposed by Bastankhah and Porté-Agel (2014) leads to:

$$C(\tilde{x}) = 1 - \sqrt{1 - \frac{C_T}{8 \left(\tilde{\sigma} + \kappa(\tilde{x})\right)^2}}.$$ (8)

To close the system, an expression of the characteristic wake width is needed. The following linear form is considered:

$$\tilde{\sigma} = \left(a_s T_i + b_s\right) \tilde{x} + c_s \sqrt{\beta},$$ (9)

with $T_i$ the turbulence intensity, $a_s$, $b_s$, $c_s$ parameters of the model and:

$$\beta = \frac{1}{2} \frac{1 + \sqrt{1 - C_T}}{\sqrt{1 - C_T}}.$$ (10)

Regarding the near-wake correction, Qian and Ishihara originally proposed a fitted form for $c_{NW}$ and $p_{NW}$. Here, a new boundary condition is introduced to determine $c_{NW}$, while $p_{NW}$ remains a parameter. Such a procedure reduces the number of constants to be calibrated. According to the actuator disk theory, the velocity at the rotor plane ($x/d_0 = 0$) is $U_d = U_\infty (1 - a)$, with $a$ the axial induction factor (see Burton et al. (2011)). The axial induction factor itself is a function of the thrust coefficient:

$$a = \frac{1}{2} \left(1 - \sqrt{1 - C_T}\right).$$ (11)

Using such a boundary condition leads to the following form:

$$c_{NW} = \sqrt{\frac{C_T}{8 \left(1 - (1 - a)^2\right)}} - c_s \sqrt{\beta}.$$ (12)

Due to the introduction of the near-wake correction, $\kappa(\tilde{x})$, Eq. (8) does not respect the mass and momentum conservation (Eq. (4)). This error is compensated by enlarging the wake: the super-Gaussian order $n$ is chosen to recover the mass and momentum conservation. Since no convenient analytical expression has been found for $n$ a priori, this is done numerically. The roots of Eq. (4) are computed, choosing $n$ as the unknown variable and for given $C(\tilde{x})$, $\tilde{\sigma}$ and $C_T$. Finally, the velocity in the wake is obtained using Eq. (1).

To sum up, the root-finding version of the super-Gaussian model is based on the following steps:

- Step 1: compute the normalized characteristic wake width using Eq. (9).

- Step 2: compute the near-wake corrected maximum velocity deficit using Eq. (8).

- Step 3: compute the super-Gaussian order $n$ using a root-finding algorithm, applied to Eq. (4).

- Step 4: compute the wake velocity using Eq. (1), and rescale using the infinite wind velocity.

### 2.2.2 Analytical approach

Numerically minimizing the mass and momentum conservation (Eq. (4)) to obtain a value for $n$, with $C(\tilde{x})$ given by Eq. (8), would lead to a strong increase in computational time. This is a major issue when dealing with wind farm design and optimization. An analytical expression for $n$ is proposed here. This expression is based on curve fitting: from the results obtained using the root-finding algorithm, it has been noticed that the evolution of $n$ against the downwind distance closely resembles to an exponential curve. The following expression is used:

$$n \approx a_f e^{b_f \tilde{x}} + c_f. \tag{13}$$

In this work, the three parameters $a_f$, $b_f$ and $c_f$ are supposed to be constants. This is a rough approximation of the super-Gaussian order $n$. It is possible to get a more precise approximation by defining the three parameters as functions of the thrust coefficient and turbulence intensity. However, this implies a larger number of parameters to be identified. The choice is made here to keep a simple form of the model and to use a limited number of parameters. The three parameters are identified based on root-finding results of Eq. (4): for a given velocity deficit (Eq. (8)), a Newton-type algorithm is used to find the value for $n$ that is a solution of Eq. (4) up to a certain tolerance. Since the root-finding problem is not so time-consuming, a large number of thrust coefficients and turbulence intensities can be considered to identify the three parameters.

The resulting analytical model is straightforward to use. Given a downstream position $\tilde{x} = x/d_0$ and a radial position $\tilde{r} = r/d_0$, a thrust coefficient $C_T$ and a turbulence intensity $T_i$, the following steps have to be followed:

- Step 1: compute the normalized characteristic wake width using Eq. (9).

- Step 2: compute the super-Gaussian order $n$ using Eq. (13).

- Step 3: compute the maximum velocity deficit using Eq. (5).

- Step 4: compute the wake velocity using Eq. (1), and rescale using the infinite wind velocity.

# 3 Calibration and Validation

## 3.1 Model calibration

The model has been calibrated using data from two experimental campaigns, thus covering a large range of turbulence intensities and thrust coefficients. The first set of data is based on Particle Image Velocimetry measurements performed in the wake of porous disks under homogeneous isotropic turbulence in a wind tunnel (see Aubrun et al. (2013) and Sumner et al. (2013)). Four cases are available and will be referred to as AD-X, X being the index of the test case. Porous disks used in these experiments are almost uniformly loaded (the disks are made of a regular metallic mesh with a larger spacing at the center): they are in accordance with the actuator disk theory used to derive the model. The second set of data is based on LiDAR measurements performed in the wake of a full-scale wind turbine. This second dataset has been used during the SWiFT benchmark, see Doubrawa et al. (2019). Details regarding this measurement campaign can be found in Herges et al. (2017). Two cases are considered, corresponding to a stable and a nearly-neutral atmosphere and will be referred to as WT-S and WT-N, S and N corresponding to the stratification (Stable or Nearly-neutral). A SOWFA (Churchfield et al. (2012)) simulation using the LES framework has been performed for the nearly-neutral case. The thrust coefficients and turbulence intensities for the six cases are summarized in Table 1.

**Table 1.** Thrust coefficients and turbulent intensities for the considered validation cases

| Case | AD-1 | AD-2 | AD-3 | AD-4 | WT-S | WT-N |
|------|------|------|------|------|------|------|
| $C_T$ | 0.43 | 0.61 | 0.56 | 0.73 | 0.75 | 0.75 |
| $T_i$ | 5% | 5% | 12% | 12% | 3.4% | 10.7% |

Based on the aforementioned six cases, the coefficients related to the wake characteristic width and the near-wake correction have been obtained, considering only the maximum of the velocity deficit at each available axial location downstream of the wind turbine. The characteristic wake width is not taken into account in this first fit. The resulting set of coefficients is given in Table 2.

**Table 2.** Fitted parameters: wake expansion and near wake correction

| $a_s$ | $b_s$ | $c_s$ | $p_{NW}$ |
|-------|-------|-------|----------|
| 0.17 | 0.005 | 0.20 | $-1$ |

The obtained parameters are different from the one proposed by other authors, such as Niayifar and Porté-Agel (2015). This may be due to the introduction of the near-wake correction in the model. Additional cases should be considered to obtain a

more robust model. It is worth noting that the parameters given in Niayifar and Porté-Agel (2015) have been obtained based on three large eddy simulations, all of them based on the same thrust coefficient ($C_T = 0.8$) with a wide range of turbulence intensities (6.9% up to 13.4% according the the presented data and the simulations described in Bastankhah and Porté-Agel (2014)). The emphasis was put on the turbulence intensity effect but not on the thrust coefficient effect, which may also explain the differences observed in the parameters.

Based on these new coefficients, another calibration procedure is applied to determine the coefficients required to obtain a value for $n$ at any given downstream location $x/d_0$, without solving the minimization problem. To get these values, a range of thrust coefficients from $0.10$ to $0.90$ and a range of turbulence intensities from $3\%$ to $20\%$ is chosen. The obtained coefficients are given in Table 3.

**Table 3.** Fitted parameters: super-Gaussian order $n$

| $a_f$ | $b_f$ | $c_f$ |
|-------|-------|-------|
| 3.11  | $-0.68$ | 2.41 |

A comparison between the proposed fit and the root-finding approach is proposed in Fig. 2. Two thrust coefficients ($C_T = 0.4$ and $C_T = 0.8$) and two turbulence intensities ($T_i = 5\%$ and $T_i = 12\%$) are considered. Despite the simplicity of the proposed expression, a reasonable agreement is observed between the proposed analytical fit and the root-finding results. The largest deviations are found in the very near wake, at downstream distances below $\tilde{x} = 2$: the maximum $\tilde{n}$ value at $\tilde{x}$ is largely underestimated.

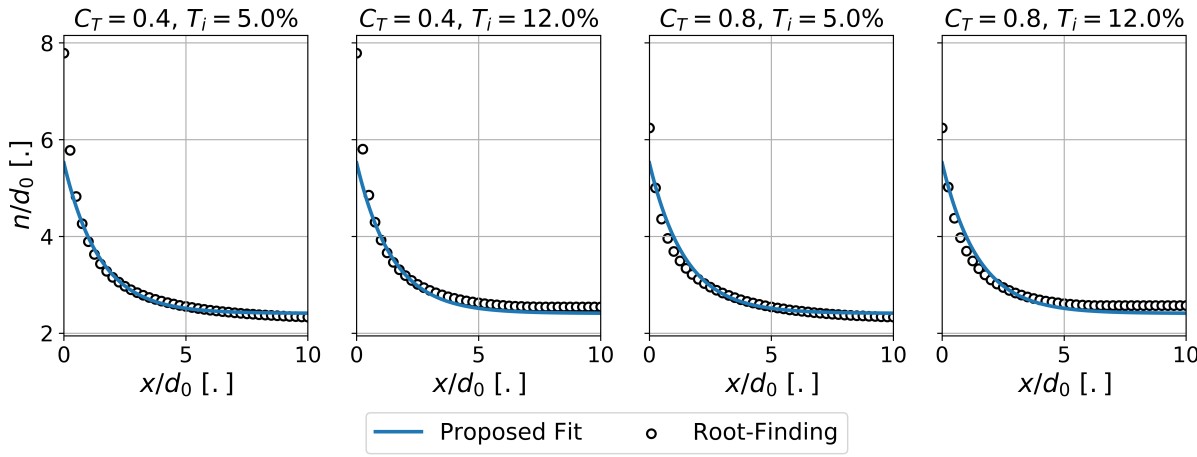

**Figure 2.** Comparison of the proposed fit for the super-Gaussian order with the root-finding approach for two thrust coefficients and two turbulence intensities.

The quality of the fit obtained using these parameters is detailed in the following subsection, based on each case used for the calibration.

## 3.2 Comparison with measured data and high-fidelity simulation

The first cases considered are the actuator disk cases. Comparisons between wake models and measurements under low turbulence conditions are given in Fig. 3. Results based on root-finding for $n$ (Eq. (4), labelled "super-Gaussian") and results based on the approximation of $n$ (Eq. (13), labelled "super-Gaussian analytical") are given. Comparisons with the Gaussian model (labelled "Gaussian") are also performed.

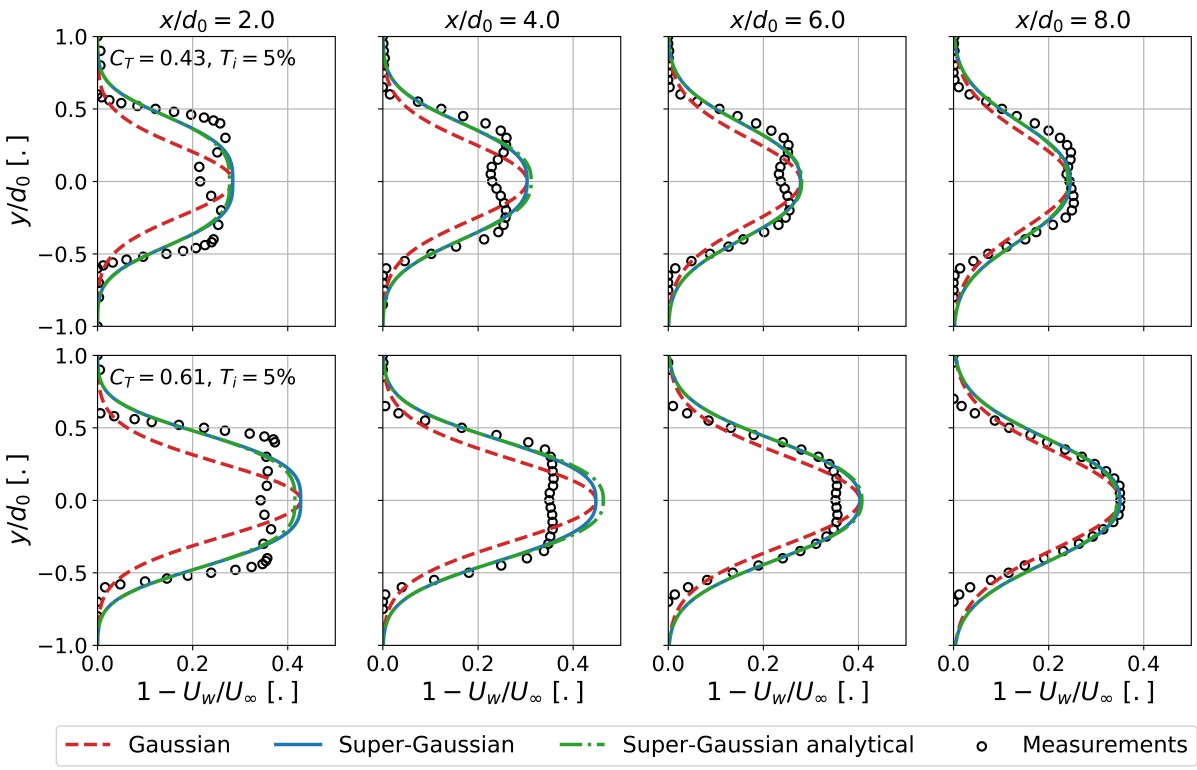

**Figure 3.** Normalized velocity deficit at four axial distances behind the actuator disk. Low turbulence case.

For both thrust coefficients, the maximum velocity deficit is slightly over-estimated, but the same trends are observed. Close to the rotor ($x/d_0 = 2$ and $x/d_0 = 4$), the velocity gradients at the edges of the wake are very strong: the wake velocity profiles tend towards a top-hat shape. In the near wake, the experimental trends are well followed, although the velocity gradients predicted by the model are not as sharp as in the measurements. At $x/d_0 = 6$, the experimental wake profile still exhibits a plateau near the center of the wake. The super-Gaussian model predicts a wider wake compared with the Gaussian model, which is consistent with the measurements.

Further downstream, at $x/d_0 = 8$, the velocity gradients are smoother, and the wake tends towards a Gaussian shape. The
wake is not fully developed, since a plateau is still observed, especially for the lowest thrust coefficient. The super-Gaussian
model reproduces this trend quite well.

At the lower thrust coefficient, the experimental data indicates a velocity decrease at the center of the wake, for all down-
stream positions. This is most probably due to the lower mesh density used at the center of the disk during the design of
the physical model (see Aubrun et al. (2013)). Differences between the analytical super-Gaussian model and the root-finding
model are almost negligible.

The impact of an higher inflow turbulence, leading to a faster wake recovery, is observed in the next cases (Fig. 4).

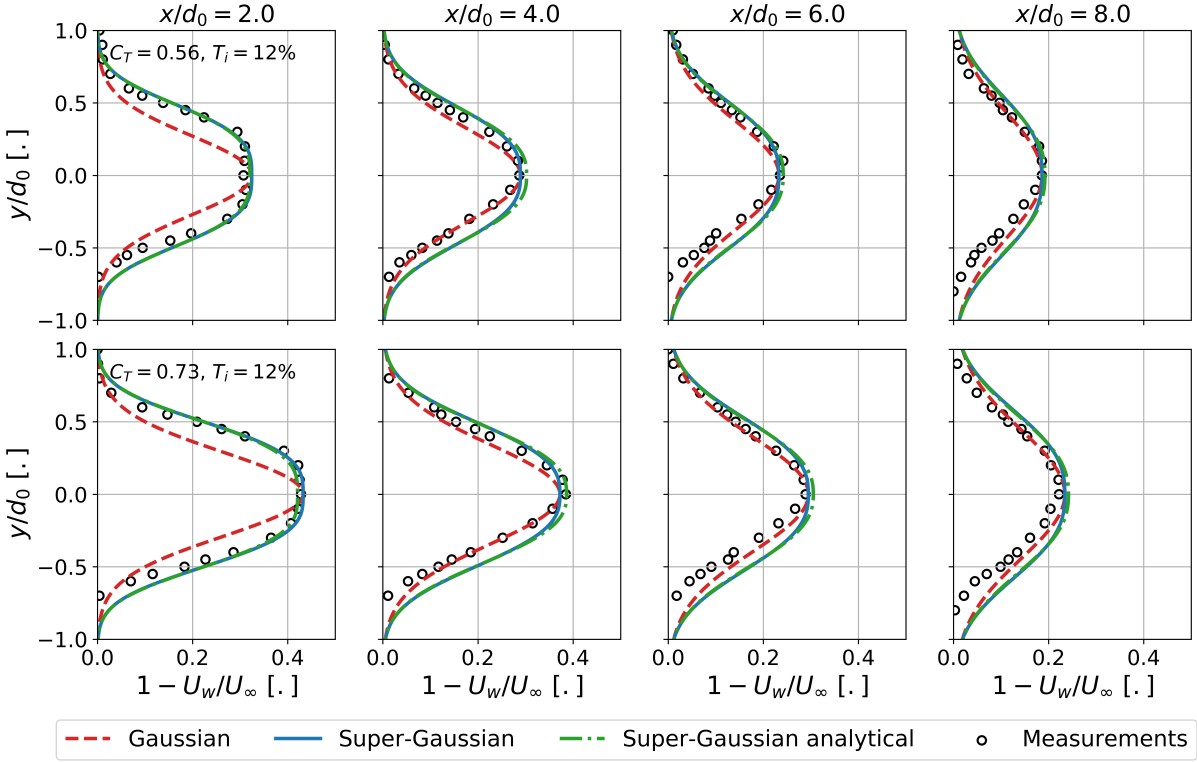

**Figure 4.** Normalized velocity deficit at four axial distances behind the actuator disk. High turbulence case.

Due to the higher turbulent level, mixing with the free flow is increased, and the plateau that was observed previously is not
present, excepted at $x/d_0 = 2$, very close to the rotor. In the near wake, at $x/d_0 = 2$ the super-Gaussian model predicts the
wake shape very well, while the Gaussian model strongly under-estimates the wake width. Downstream, at $x/d_0 = 4$, 6 and
8, the wake width is slightly over-estimated by the super-Gaussian model. The Gaussian model is more in agreement with the
measurements, but differences are small. Again, differences between the analytical model and the root-finding counterpart are
very small, even negligible.

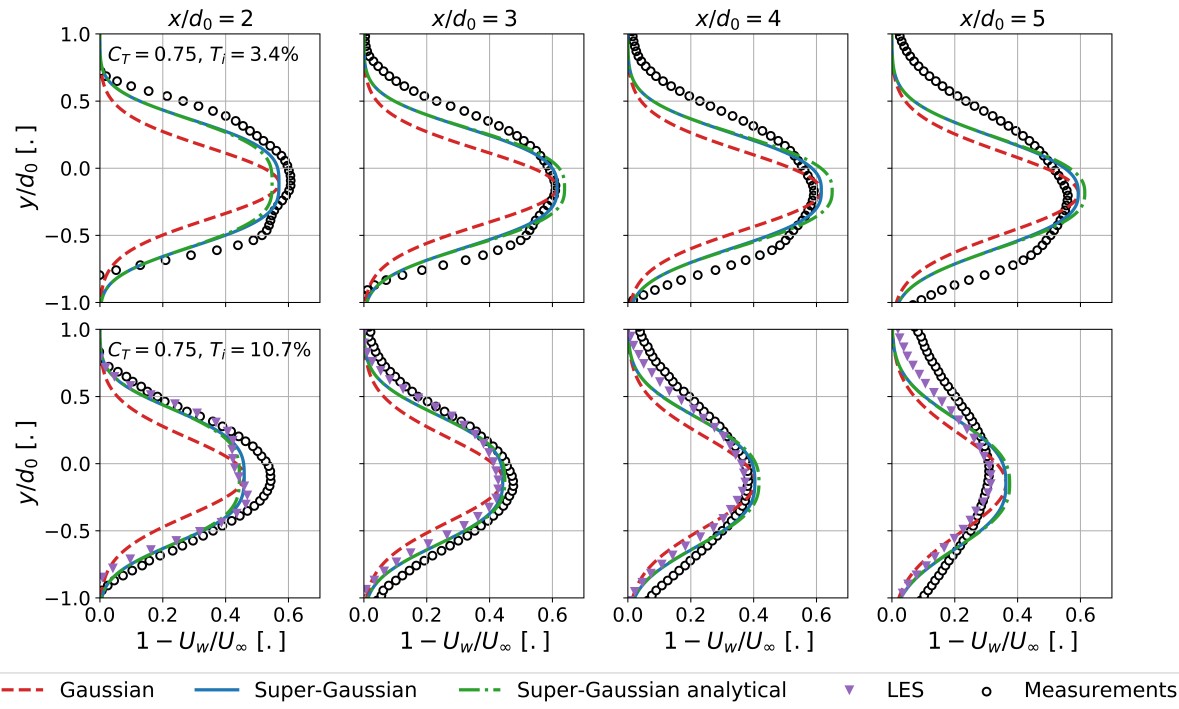

**Figure 5.** Normalized velocity deficit at four axial distances behind the wind turbine. Stable case (WT-S), top, and nearly-neutral case (WT-N), bottom.

Last, comparisons are made between the wake models and the SWiFT measurements. A stable, low turbulence case and a nearly-neutral, higher turbulence case are presented in Fig. 5, at the top and bottom, respectively. For the nearly-neutral case, results from a LES simulation, based on the SOWFA library (Churchfield et al. (2012)), are also included. A slight offset has been imposed in the $y/d_0$ direction for all simulations, including the LES, to compensate for the wake deflection observed in the measurements. Measurements also reveal a slight asymmetry in the wake velocity profile, that is not accounted for in the analytical models. In terms of maximum velocity deficit, the agreement between the wake models and measurements is good, despite a slight under-estimation of the velocity deficit at $x/d_0 = 2$ and a slight over-estimation at $x/d_0 = 5$ for both stable and nearly-neutral cases. The LES results also slightly under-estimate the velocity deficit at $x/d_0 = 2$. In terms of wake shapes, the super-Gaussian model predicts wider wakes than the Gaussian model, as expected, and is more in line with the measurements. A good agreement is observed between the super-Gaussian model and the LES simulations, despite some differences at $x/d_0 = 5$. The LES predicts a slightly thinner wake compared with the measurements. The super-Gaussian model clearly improves the wake shape prediction. Some differences appear between the analytical super-Gaussian model and the root-finding version, the root-finding version being closer to the experiment. The maximum velocity deficit at $x/d_0 = 3, 4$ and $5$ is slightly over-estimated for the stable case.

For a more quantitative comparison, the normalized $L2$ error between each model and the experimental velocity deficit are provided in Fig. 6.

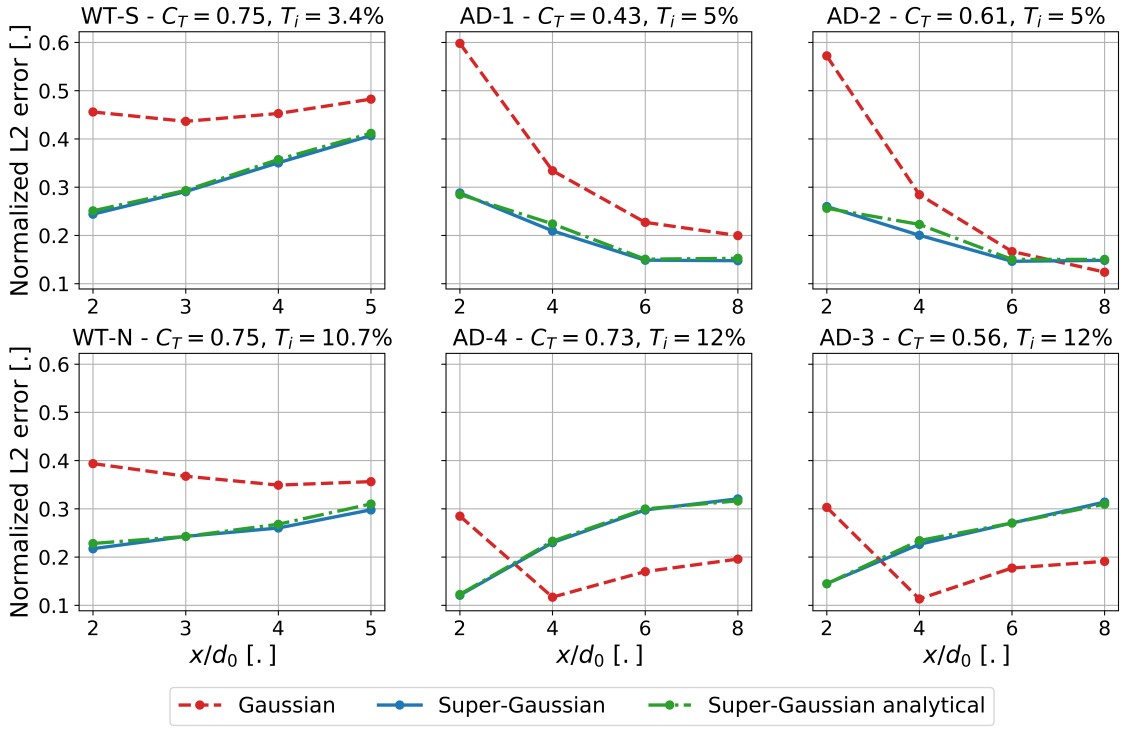

**Figure 6.** Normalized $L2$ error between the wake models and measured velocity deficit for the six considered cases. Lower inflow turbulence cases on the top line, higher inflow turbulence cases on the bottom line.

Results for the lower inflow turbulence are on the top, and higher inflow turbulence cases on the bottom of the figure. For the wind turbine case (left plots), results are very satisfactory, since the error is lowered at all downstream positions. The improvement is more pronounced in the near wake: the difference between the super-Gaussian and the Gaussian models, in terms of error, tends to diminish with the distance to the rotor. This is the case for both low inflow turbulence and high inflow turbulence cases. The Gaussian and the super-Gaussian model being based on the same maximum velocity deficit models, the improvement is due to the enlarged wake that is obtained using the super-Gaussian model. The wake model predictions are improved with the super-Gaussian model up to five diameters behind the wind turbine for the WT-S and WT-N cases, which is a separation distance that is commonly observed in offshore or onshore wind farm layouts. This highlights the usefulness of the super-Gaussian model for wind farm design purposes.

If the super-Gaussian model clearly improves the results for the wind turbine cases, results are less satisfactory for the actuator disk cases under high inflow turbulence. A clear improvement is observed for the low ambient turbulence conditions: for normalized distance to the rotor plane of two to six disk diameters, the normalized $L2$ error is lower with the super-Gaussian

model compared to the Gaussian model. Again, the impact is more pronounced in the near-wake and tends to diminish in the far wake, which is expected, since the super-Gaussian shape function tends to the Gaussian shape in the far wake. In the far wake, the Gaussian model has a lower error than the super-Gaussian model at $x/d_0 = 8$ for the higher $C_T$ case. Looking at the wake velocity profiles (Fig. 3), the higher error observed with the super-Gaussian model can be attributed to an overestimation of the wake width. At this location, the value of the super-Gaussian order, $n$, is not equal to two, and the Gaussian model better predicts the wake shape. This is also observed for the high inflow turbulence case. For both thrust coefficients, the super-Gaussian model lowers the error in the near-wake ($x/d_0 = 2$), but increases the error at the other positions (i.e. in the far wake). In order to recover the accuracy of the Gaussian model, the super-Gaussian order $n$ should be equal to two for these cases. A practical way to improve the super-Gaussian model is to find a better calibration for the near-wake correction, Eq. (7) and/or the characteristic wake width, Eq. (9): since the near-wake power coefficient, $p_{NW}$, has a rather low value in the proposed calibration, the near-wake correction has an impact in the far wake that might be over-estimated, leading to super-Gaussian order values that are above two. This highlights the difficulty to properly calibrate analytical wake models, and the need for more measurements and high-fidelity simulations. Nevertheless, no explanations has be found to justify the differences observed between the wind turbine case WT-N and the actuator disk case AD-4: operating conditions are similar in terms of thrust coefficient and turbulence intensities, but a larger wake is observed in the wind turbine case, which leads to different conclusions in terms of super-Gaussian model performance compared with the Gaussian model. For all the considered cases, the normalized $L2$ errors for the super-Gaussian model based on the root-finding algorithm and the analytical one are very similar. No noticeable differences are observed, excepted for the AD-2 case at $x/d_0 = 4$, for which a slight overestimation of the maximum velocity deficit was observed in Fig. 3.

## 4 Conclusions

A super-Gaussian model for wind turbine wakes has been introduced. The model transitions from a nearly top-hat shape in the near wake to the well-known Gaussian shape in the far wake. The super-Gaussian order, $n$, which determines the shape of the wake, is deduced by finding the root of the mass and momentum conservation equation. To avoid the numerical evaluation of the root-finding problem and save computational time, a simple analytical expression for the super-Gaussian order $n$ has been proposed. Comparisons with wind tunnel, Particle Image Velocimetry measurements behind a porous disk and LiDAR measurements in the wake of a full-scale wind turbine show a good agreement between the model and the measured data. While the comparisons show an improvement compared with the Gaussian model, there are still large differences between the model predictions and measurements or LES simulations, highlighting the need for a more extensive calibration of the model. In the near wake, the model also compares well with a LES simulation. The model improves the Gaussian model by predicting an enlarged wake, consistent with observations, even at distances down to six diameters behind the wind turbine. Future work should include an extensive calibration and validation of the model, considering additional turbulence intensities and thrust coefficients. A model for the hub wake could also be integrated in the model. Comparisons at the wind farm scale are also planned in a near future. This implies the use of a wake-added turbulence model. In the super-Gaussian model, $n$ can

be considered as an implicit representation of the shear layer that expands downstream the wind turbine, starting near the tip

of the blades (Sanderse (2009)). There is most probably a link to consider between the super-Gaussian model and wake-added

turbulence models.

*Code and data availability.* A python implementation of the analytical super-Gaussian model as well as numerical results can be made available upon request from the corresponding author. An implementation of this super-Gaussian model has been undertaken by the NREL in the FLORIS solver (https://github.com/NREL/floris).

## Appendix A: Detailed derivation of the super-Gaussian model

According to Frandsen et al. (2006), the application of mass and momentum conservation leads to the following expression:

$$\rho \int_0^\infty U_w(U_\infty - U_w)dA = T. \tag{A1}$$

From Eq. (3), Eq. (A1) writes:

$$\int_0^\infty U_w(U_\infty - U_w)rdr = \frac{C_T U_\infty^2 d_0^2}{16}. \tag{A2}$$

Introducing the normalized radius, $\tilde{r} = r/d_0$, Eq. (A2) becomes:

$$\int_0^\infty U_w(U_\infty - U_w)\tilde{r}d\tilde{r} = \frac{C_T U_\infty^2}{16}. \tag{A3}$$

Inserting the super-Gaussian shape function and using $C(\tilde{x})$, the maximum velocity deficit (Eq. (1)) leads to:

$$\int_0^\infty U_w(U_\infty - U_w)d\tilde{A} = \int_0^\infty \left( U_\infty C(\tilde{x})e^{-\frac{\tilde{r}^n}{2\tilde{\sigma}^2}} \times U_\infty \left( 1 - C(\tilde{x})e^{-\frac{\tilde{r}^n}{2\tilde{\sigma}^2}} \right) \right) \tilde{r}d\tilde{r}$$
$$= U_\infty^2 C(\tilde{x}) \int_0^\infty \left( e^{-\frac{\tilde{r}^n}{2\tilde{\sigma}^2}} - C(\tilde{x})e^{-\frac{\tilde{r}^n}{\tilde{\sigma}^2}} \right) \tilde{r}d\tilde{r}. \tag{A4}$$

A known form for the primitive of $xe^{cx^n}$ exists:

$$\int xe^{cx^n}dx = -\frac{\Gamma_i(2/n, -cx^n)x^2}{n(-cx^n)^{2/n}}, \tag{A5}$$

$\Gamma_i(n, x)$ being the upper incomplete Gamma function. Fortunately, this form has finite limits at both infinity and positive zero. These limits write:

$$\lim_{x \to \infty} \left( -\frac{\Gamma(2/n, -cx^n), x^2}{n(-cx^n)^{2/n}} \right) = 0, \qquad \lim_{x \to 0^+} \left( -\frac{\Gamma(2/n, -cx^n)x^2}{n(-cx^n)^{2/n}} \right) = -\frac{\Gamma(2/n)}{n(-c)^{2/n}}, \tag{A6}$$

with $\Gamma(x)$ the Gamma function. Inserting (A6) into (A4) and choosing the correct form for $c$ leads to the following form:

$$\int_0^\infty U_w(U_\infty - U_w)d\tilde{A} = -U_\infty^2 C(\tilde{x})\Gamma(2/n)\left(\frac{C(\tilde{x})}{n}\tilde{\sigma}^{4/n} - \frac{2^{2/n}}{n}\tilde{\sigma}^{4/n}\right). \tag{A7}$$

Inserting (A7) into (A3) leads to:

$$\tilde{\sigma}^{4/n}C(\tilde{x})^2 - 2^{2/n}\tilde{\sigma}^{4/n}C(\tilde{x}) + n\frac{C_T}{16\Gamma(2/n)} = 0. \tag{A8}$$

Considering $C(\tilde{x})$ as the variable to solve for, Eq. (A8) is a quadratic expression of second degree for which solutions are well-known. The discriminant is given by:

$$\Delta = \left(2^{2/n}\tilde{\sigma}^{4/n}\right)^2 - \frac{n\tilde{\sigma}^{4/n}C_T}{4\Gamma(2/n)}. \tag{A9}$$

Finally, the roots of the polynomial expression are obtained:

$$C(\tilde{x}) = \frac{2^{2/n}\tilde{\sigma}^{4/n} \pm \sqrt{\Delta}}{2\tilde{\sigma}^{4/n}} = 2^{2/n-1} \pm \sqrt{2^{4/n-2} - \frac{nC_T}{16\Gamma(2/n)\tilde{\sigma}^{4/n}}}. \tag{A10}$$

As for the Gaussian model, only solutions based on the minus sign lead to physical solutions for the velocity deficit. The final form of the maximum velocity deficit is:

$$C(\tilde{x}) = 2^{2/n-1} - \sqrt{2^{4/n-2} - \frac{nC_T}{16\Gamma(2/n)\tilde{\sigma}^{4/n}}}. \tag{A11}$$

*Author contributions.* The first author derived the super-Gaussian wake model, performed the model fitting and the validation. The second author performed the LES simulation and participated in the model derivation and implementation. Both authors contributed to the final manuscript.

*Competing interests.* The authors declare no conflict of interest.

*Acknowledgements.* The authors are grateful to Sandrine Aubrun for sharing the porous disk database, and to the partners involved in the SWiFT experimental campaign. We also would like to thank Paul Fleming and his colleagues from the NREL for considering the implementation of the super-Gaussian model in the FLORIS solver.

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
