# Peer review of "An alternative form of the super-Gaussian wind turbine wake model"

_Wind Energy Science, 2019_

## Referee Comment (RC1) · Marijn Floris van Dooren (Referee) · 26 Mar 2020

This paper presents a clear and well documented analysis of a super-Gaussian model for describing wind turbine wakes, especially trying to improve the accuracy w.r.t. the standard single-Gaussian wake model in the near wake. I think the paper is well written and feels very mature. However, there are still some weak spots that could be improved. Therefore I suggest a minor revision.

General comments:

- The paper highlights the differences between the super-Gaussian and single-Gaussian wake shape very well. However, the paper does not address another common way to describe especially the near wake behavior, which is the double-

Gaussian shape. Recent examples on this are:

https://doi.org/10.5194/wes-5-237-2020

https://doi.org/10.1088/1742-6596/753/3/032039

It would be appropriate if the authors also mentioned this model and include a comparison to this particular approach in their work. For example in Figure 2 the double-Gaussian wake shape can be recognized for the actuator disk wake.

- Since the paper puts a clear emphasis on defining $n(x)$ as the nth order of a variable super-Gaussian shape, it would be very interesting to have a visualization of $n(x)$ for one or more cases.

- You conclude that the highlighted cases show a good agreement between the model and the measurements. I agree, you clearly demonstrate that your model approximates the wake shape better than the single-Gaussian does it. However, in most cases there is still quite a large error remaining. The conclusion on the performance might be formulated in a slightly more critical way.

Specific comments:

- P4, L83: The thrust coefficient is described as manufacturer data. While I agree that it is a design parameter, I suggest to elaborate more on the physical meaning of it and whether you consider it as a constant or a variable in your analysis.

- P4, L97: You state that 'an unknown variable, the order of the super-Gaussian n appears'. I would phrase it differently, since n was already introduced on P2. For the reader it is not unknown anymore.

- P7, L170: I assume you use a fit of Eq. (9) to determine the parameters as, bs and cs? I do not understand how the standard deviation can be omitted in this process. Could you elaborate on this? Also it would be nice to have some

more information about the quality of the fit, maybe addressing the residuals or visualizing the fit.

- P7, L172: Related to the previous comment, how big was the data set that Niayifar and Porté-Agel (2015) used to determine their set of parameters? Maybe this can also (partly) explain the differences.

Technical corrections:

- Because there is only one author affiliation, the footnote notation using the number 1 is not necessary.

- P1, L3: Consider replacing 'made on' with 'of'.

- P1, L16/23/24: Consider replacing 'inter-distance' with 'separation distance'.

- P2, L26: I recommend to change 'two shortcomings, that are actually closely related, need to be alleviated' to 'two closely related shortcomings have to be alleviated'.

- P2, L31: Consider replacing 'evolves' with 'evolve', since the subject (wake velocity profiles) is plural.

- P6, L135: Consider replacing 'Minimizing numerically' with 'Numerically minimizing'.

- P6, L139: Consider replacing 'follows more or less' with 'closely resembles'.

- P6, L145; Consider replacing 'solution' with 'a solution'.

- P11, L230: Consider replacing 'inter-distance' with 'separation distance'.

- P13, L275: The brackets around 'A2' are missing.

---

## Referee Comment (RC2) · Anonymous Referee #2 · 12 Jun 2020

Thank you for this paper. I found it to be a very clear and well-constructed paper. The super-Gaussian model presented satisfies a clear need for an improved near wake model for engineering models of the wake. The paper therefore makes an important and valuable contribution to highly active fields of research include wind farm modeling and control. The introduction covers the literature and clearly explains the location and contribution of the present work. Finally, the paper's good writing, clear figures, and well-explained formulas make it very direct to comprehend and understand.

A general comment could be it would be interesting to see how this new model compares to pre-existing models in comparisons of wind farm SCADA data, particularly for farms with near-wake conditions, such as you mention Lilligrund. This could perfectly well be a follow-up paper, but if you've made any checks would be interesting to

mention.

Excellent contribution and only one very minor comment is from section 2.1

2.1: " CT , which is a manufacturer data." – recommend a better explanation of Ct represents

---

## Author Comment (AC1) · 9 Jul 2020

**Reviewer 1:**

This paper presents a clear and well documented analysis of a super-Gaussian model for describing wind turbine wakes, especially trying to improve the accuracy w.r.t. the standard single-Gaussian wake model in the near wake. I think the paper is well written and feels very mature. However, there are still some weak spots that could be improved. Therefore I suggest a minor revision.

General comments:

- The paper highlights the differences between the super-Gaussian and single-Gaussian wake shape very well. However, the paper does not address another common way to describe especially the near wake behavior, which is the double-Gaussian shape. Recent examples on this are:
  - https://doi.org/10.5194/wes-5-237-2020
  - https://doi.org/10.1088/1742-6596/753/3/032039

  It would be appropriate if the authors also mentioned this model and include a comparison to this particular approach in their work. For example in Figure 2 the double-Gaussian wake shape can be recognized for the actuator disk wake.
  - We agree with you that a comparison between the super-Gaussian and the double-Gaussian models would be interesting.
    Indeed, a "double-Gaussian" wake shape can be observed in the near wake in Figure 3 (formerly Figure 2). Nevertheless, as discussed in Aubrun et al. paper, the measured "double-Gaussian" wake shape is attributed to the lower mesh density at the center of the porous disk used during the experiments. As we are not sure that the mesh properties of the porous disk are fully representative of hub/nacelle effects, we decided that we will not take into account this near-wake characteristics on this specific test case in a first step.
    Comparisons between our proposed model and the double-Gaussian wake models are not straightforward, since, to our best knowledge, no "generic" formulation (or calibration) of the double-Gaussian wake model has been proposed yet (i.e. no dependence on thrust coefficient or turbulence intensities are considered in the two aforementioned references).
  - Furthermore, the analysis of some experimental data (i.e. EPFL measurements) shows that, in the very near wake, the hub and nacelle effect implies an additional velocity deficit, and not a reduction of the velocity deficit at the wake center, as it is proposed in the double-Gaussian model. Finally, the "zero-deficit" obtained at the wake center at the rotor plane using the double-Gaussian model (see Figure 4 in https://doi.org/10.1088/1742-6596/753/3/032039) seems rather unphysical.
  - From these arguments, it is our interpretation that the hub/nacelle effect and potential other sources of non-uniformities in the near wake should be treated separately: one can adopt the super-Gaussian shape, and further correct the velocity deficit to account for the hub/nacelle effects of non-uniform induction distributions at the rotor plane. This is briefly discussed lines 53 to 58.

- Since the paper puts a clear emphasis on defining n(x) as the nth order of a variable super-Gaussian shape, it would be very interesting to have a visualization of n(x) for one or more cases.
    - Such visualization have been added (fig. 2), including 2 turbulence intensities and 2 thrust coefficients.

- You conclude that the highlighted cases show a good agreement between the model and the measurements. I agree, you clearly demonstrate that your model approximates the wake shape better than the single-Gaussian does it. However, in most cases there is still quite a large error remaining. The conclusion on the performance might be formulated in a slightly more critical way.
    - Indeed there is still a large error between the model and the experimental measurements. The following sentence has been added in the conclusion:
        - "While the comparisons show an improvement compared with the Gaussian model, there are still large differences between the model predictions and measurements or LES simulations, highlighting the need for a more extensive calibration of the model."

Specific comments:

- P4, L83: The thrust coefficient is described as manufacturer data. While I agree that it is a design parameter, I suggest to elaborate more on the physical meaning of it and whether you consider it as a constant or a variable in your analysis.
    - In order to clarify the physical meaning of the thrust coefficient, the following sentence has been added:
        - "As shown by Eq. 2 and 3, the thrust coefficient is a non-dimensional variable, that represents the loss of kinetic energy of the flow due to the presence of the wind turbine. It scales the intensity of the velocity deficit in the wake"
    - The dependence of the thrust coefficient on infinite wind velocity has been clarified:
        - " supposed to be known as a function of the infinite wind velocity in the rest"

- P4, L97: You state that 'an unknown variable, the order of the super-Gaussian n appears'. I would phrase it differently, since n was already introduced on P2. For the reader it is not unknown anymore.
    - This has been rephrased:
        - Line 100: "another variable is introduced: the super-Gaussian order n."

- P7, L170: I assume you use a fit of Eq. (9) to determine the parameters as, bs and cs? I do not understand how the standard deviation can be omitted in this process. Could you elaborate on this? Also it would be nice to have some more information about the quality of the fit, maybe addressing the residuals or visualizing the fit.
    - Indeed, these three parameters have been calibrated to obtain a first, acceptable agreement to experimental data. These parameters are used to estimate the characteristic wake width, given a thrust coefficient and a turbulence intensity. They have been obtained using the maximum velocity deficit (experimental one) and

trying to match it using the proposed analytical formulation. The presented coefficients are a first estimate, used in the present study. Further work has been made on the model calibration, using an LES solver, and will be presented in a future publication.

- P7, L172: Related to the previous comment, how big was the data set that Niayifar and Porté-Agel (2015) used to determine their set of parameters? Maybe this can also (partly) explain the differences.
  - o Niayifar and Porté-Agel used a set of three LES simulations, using one single thrust coefficient of 0.8. Indeed, this may explain some noticed differences. To make it clear, a discussion has been integrated (L187 to L191).

Technical corrections:

- Because there is only one author affiliation, the footnote notation using the number 1 is not necessary.
- P1, L3: Consider replacing 'made on' with 'of'.
- P1, L16/23/24: Consider replacing 'inter-distance' with 'separation distance'.
- P2, L26: I recommend to change 'two shortcomings, that are actually closely related, need to be alleviated' to 'two closely related shortcomings have to be alleviated'.
- P2, L31: Consider replacing 'evolves' with 'evolve', since the subject (wake velocity profiles) is plural.
- P6, L135: Consider replacing 'Minimizing numerically' with 'Numerically minimizing'.
- P6, L139: Consider replacing 'follows more or less' with 'closely resembles'.
- P6, L145; Consider replacing 'solution' with 'a solution'.
- P11, L230: Consider replacing 'inter-distance' with 'separation distance'.
- P13, L275: The brackets around 'A2' are missing.

  - o Thank you for these technical corrections, that have all been integrated in the text.

**Reviewer 2:**

Thank you for this paper. I found it to be a very clear and well-constructed paper. The super-Gaussian model presented satisfies a clear need for an improved near wake model for engineering models of the wake. The paper therefore makes an important and valuable contribution to highly active fields of research include wind farm modeling and control. The introduction covers the literature and clearly explains the location and contribution of the present work. Finally, the paper's good writing, clear figures, and well-explained formulas make it very direct to comprehend and understand.

- o Thank you very much for these positive feedbacks!

A general comment could be it would be interesting to see how this new model compares to pre-existing models in comparisons of wind farm SCADA data, particularly for farms with near-wake conditions, such as you mention Lilligrund. This could perfectly well be a follow-up paper, but if you've made any checks would be interesting to mention.

- o Indeed, comparison with full scale wind farm SCADA would be interesting. It has not been integrated to this paper, since it requires to define additional models, such as superposition strategy, added turbulence models, etc. As you mention, we thought it could be part of a follow-up paper, in order not to over-charge the present communication. However, researchers from the NREL already analyzed the impact of the super-Gaussian model on the power production of downstream wind turbines, and will probably communicate on these results.

Excellent contribution and only one very minor comment is from section 2.1 2.1: " CT , which is a manufacturer data." – recommend a better explanation of Ct represents

- o Some details regarding the thrust coefficient have been added in the paper:
  - "As shown by Eq. 2 and 3, the thrust coefficient is a non-dimensional variable, that represents the loss of kinetic energy of the flow due to the presence of the wind turbine. It scales the intensity of the velocity deficit in the wake."

---

## Author Response (AR2)

Answer to Referee #1

- The formatting of the variables CT and Ti in Figure 2 should match Figure 6.

      -> The figure has been changed.

- When referring to Equation (2) and (3), brackets around the numbers are missing in line 101.

      -> The brackets have been added.

Answer to Referee #2

None.

[revised manuscript text omitted]